



# Shipborne Comparison of Infrared and Passive Microwave Radiometers for Sea Surface Temperature Observations

Guisella Gacitúa[1], Jacob L. Høyer[1], Sten Schmidl Søbjærg[2], Hoyeon Shi[1], Sotirios Skarpalezos[1], Ioanna Karagali[1], Emy Alerskans[1], and Craig Donlon[3]

[1]Danish Meteorological Institute, Sankt Kjelds Plads 11, 2100 Copenhagen, Denmark
[2]DTU-Space, Technical University of Denmark, Elektrovej 327, 2800 Kongens Lyngby, Denmark
[3]European Space Agency/ESTEC, Keplerlaan 1, 2201 AZ Noordwijk, The Netherlands

**Correspondence:** Guisella Gacitúa (gga@dmi.dk)

**Abstract.** In the spring of 2021, a shipborne comparison of sea surface temperature (SST) measurements was undertaken using Thermal Infrared (TIR) and Passive Microwave (PMW) radiometers. The Danish Meteorological Institute (DMI) and the Technical University of Denmark (DTU) jointly deployed two TIR and two PMW instruments aboard the Norröna ferry, which traversed between Denmark and Iceland for a week. The primary objective was to assess the proximity-based comparison of
TIR and PMW measurements, minimizing atmospheric influences and providing valuable insights into skin (TIR) and sub-skin (PMW) SSTs. A linear regression algorithm was developed using TIR SST data as a reference to derive PMW SST from brightness temperature. The data analysis primarily focused on evaluating data variability, identifying discrepancies between TIR and PMW SST, and assessing the overall uncertainty in the retrieval process. The overall root mean squared error (RMSE) of the retrieved PMW SST was 0.88 K during the ship's motion and 0.94 K under stable conditions when the ship was
moored. The analysis of the retrieved SST error budget involved the consideration of observed quantities and a forward model, accounting for factors like instrument noise, wind speed, incident angles, and the RMSE of skin and sub-skin temperature. The resulting error budget indicated 0.97 K for the data acquired during motion and 0.34 K for data collected during port stay.

## 1 Introduction

Sea surface temperature (SST) is a fundamental variable to observe and is recognized as an essential climate variable (ECV)
(Bojinski et al., 2014). SST regulates ocean-atmosphere interactions and plays a crucial role as a significant input in atmospheric and oceanic forecasting models. In addition, the assessment of climate change and variability heavily relies on remote sensing-based observations of SST, which have been collected for over five decades, resulting in a substantial and extensive dataset (Minnett et al., 2019; Merchant et al., 2019). The most extensive satellite records providing global coverage of SST have traditionally been acquired through the use of Thermal Infrared (TIR) satellite sensors that measure the radiation emitted
from the skin of the sea surface (Donlon et al., 2007). Such sensors have been available since the early 1980s and have a typical spatial resolution of 1-4 km and uncertainties of about 0.2-0.4°C (e.g. Embury et al., 2012; Gladkova et al., 2016). TIR satellite SST observations are thus very accurate yet are subject to certain limitations, e.g. can only be obtained in cloud-free conditions and are influenced by the presence of aerosols and water vapor.



An alternative method for retrieving SST involves utilizing Passive Microwave (PMW) satellite measurements of brightness temperature ($T_b$) in C- and X- bands that is emitted from the sub-skin layer of the ocean surface (Gentemann et al., 2010). PMW sensors have been available since 1997 and can provide observations of the sea surface in non-precipitating conditions. The quality of the PMW observations is impacted by strong winds (rough sea state), sun-glint, and Radio Frequency Interference (RFI). In addition, proximity to land and sea ice (within ∼100 km), can contaminate observations of the sea surface (Gentemann, 2014; Gentemann and Hilburn, 2015). PMW SST products typically have uncertainties of 0.4-0.5°C with a spatial resolution of 50-60 km (Alerskans et al., 2020; Nielsen-Englyst et al., 2018; Gentemann, 2014).

As discussed in O'carroll et al. (2019), it is vital that the satellite constellation consists of both TIR and PMW satellite sensors, as these two types of sensors have complementary observational characteristics but represent two different physical observations such as the temperature of the skin (TIR) and subskin (PMW) surface layer and differ by the cool skin effect (Donlon et al., 2002). Conversely, studies comparing TIR and PMW satellite observations of SST have shown significant discrepancies over large regions and on monthly time scales (Castro et al., 2008; Gentemann, 2014). Due to their different observational characteristics, it is important to link PMW and TIR SST observations and to quantify the different contributions to potential discrepancies between TIR and PMW SST. This is particularly important when generating consistent climate data records and is supported by the current EU Copernicus plans calling for an improved understanding of TIR and PMW SSTs; the development of the new Copernicus Imaging Microwave Radiometer (CIMR) that will ensure the acquisition of accurate and high-resolution PMW observations in parallel with the Sentinel 3 TIR SST observations for many years (Thépaut et al., 2018; Jiménez et al., 2021; Nielsen-Englyst et al., 2021).

Fiducial Reference Measurements (FRMs) have been identified as essential observations for the validation and improvement of the satellite SST products (Donlon et al., 2014b; O'carroll et al., 2019; Le Menn et al., 2019). Existing projects such as SHIPS4SST (ships4sst.org) are ongoing and collecting SST FRM from e.g. TIR radiometers to be used for satellite validations. Laboratory and inter-comparison campaigns have been conducted to assess the performance of the System of Units (SI) traceable FRM TIR radiometers (Wimmer et al., 2012; Theocharous et al., 2010, 2019). The collection and deployment of PMW radiometers on ships to observe the sea surface temperature are, however, more complex and less mature compared to TIR radiometers, and as a result, very few coinciding microwave and TIR radiometric observations of the sea surface temperature are available.

This study presents the inter-comparison of TIR and PMW radiometer instruments for the measurement of SST collected during a shipborne campaign. This experiment is the first of its kind using FRM TIR instruments. It offered a unique opportunity to simultaneously observe and compare TIR and PMW measurements in close proximity to the sea surface, thereby minimizing the potential influence of atmospheric factors on the collected data. The primary objectives of this investigation are to gain experience with shipborne PMW deployments and to enhance our understanding of the relationship between SST at the skin and sub-skin levels as observed by the two types of radiometers. The analysis focuses on i) Quantifying the instrumental noise and geophysical variability of brightness temperature ($T_b$) PMW data collected during the experiment, ii) Assessing the geophysical conditions contributing to the variability of the observed PMW data, iii) Retrieving $SST_{subskin}$ from (PMW) using





a statistical model, iv) Quantifying the error budget of the retrieved PMW SSTs and v) Analysing the differences between the retrieved $SST_{subskin}$ and $SST_{skin}$, as well as against existing satellite products.

The results provide insights for improving upcoming inter-comparison campaigns, helping establish connections between these two measurement techniques and optimizing the current synergy between TIR and PMW radiometers.

## 2    Data and Methodology

### 2.1    TIR Instrument - ISAR

The infrared SST autonomous radiometer (ISAR) is specifically designed for shipborne measurements of SST at the skin

interface. Over the course of nearly two decades, ISARs have proven to be highly effective in collecting accurate SST data from ships. These instruments are commonly deployed for data validation purposes, particularly in the collection of FRM used to validate satellite-derived SST data (Donlon et al., 2008, 2014b; Wimmer and Robinson, 2016).

ISARs utilize a Heitronics KT15.85D infrared detector and are equipped with two precision calibration blackbodies (BBs). One BB is maintained at the ambient temperature, while the other is heated to approximately 12 K above ambient. The scanning

process of the ISAR involves a sequential set of observations. Initially, the infrared detector points towards the calibration blackbodies, allowing for initial calibration. Subsequently, the detector scans the sky and the sea, which serves as a self-calibration reference. This comprehensive scanning process enables the ISAR to achieve a remarkable level of accuracy, with an error range of 0.1 K Root Mean Square Error (RMSE) (Donlon et al., 2008; Wimmer and Robinson, 2016).

To ensure data integrity, the ISAR system incorporates a rain detector mechanism that effectively prevents water intrusion.

As a result, the instrument stops obtaining sea measurements during rainy conditions.

### 2.2    PMW Instrument - EMIRAD

The EMIRAD radiometers, owned and operated by the Technical University of Denmark (DTU) - Space, underwent special refurbishment for the purpose of conducting the TIR-PMW inter-comparison experiment. The refurbished EMIRAD-C and EMIRAD-X models utilize horn antennas, connected to the receiver via an Ortho Mode Transducer (OMT), which enables the

independent output of signals for the two polarizations through separate connector ports (Høyer et al., 2021). EMIRAD-C is fully polarimetric and capable of simultaneously measuring the complete Stokes vector in the C band. EMIRAD-X measures the two polarizations in a time multiplex using the same physical receiver in the X band. Frequencies of C and X band radiometers are highly advantageous for deriving and calibrating PMW SST products. These frequencies play a central role in accurately measuring surface temperature, as highlighted by previous studies (Nielsen-Englyst et al., 2021; Prigent et al., 2013).

Especially for the C band frequency of 7.05 GHz (see Table 1), which is very close to the frequency of the first channel (6.925 GHz) of the Advanced Microwave Scanning Radiometer (AMSR), sensitivity in cold waters is higher (Wentz and Meissner, 2000), highly relevant for the area of the field campaign. In order to achieve optimal consistency with satellite observations,





an average incident angle of 55 degrees was selected, aligning closely with the AMSR for EOS (AMSR-E) and AMSR2 (Alerskans et al., 2020; Mai et al., 2016).

The calibration procedure for EMIRAD involves a series of four steps, which encompass internal calibration as well as additional corrections to account for instrument-related effects and the antenna's attitude (Søbjærg et al., 2013, 2015).

**Table 1.** General characteristics of the radiometers used for this shipborne inter-comparison campaign.

| Qty. | Radiometer type | Name | Wavelength $\mu$m | Frequency GHz | Bandwidth | Sea-view angle |
|------|-----------------|------|-------------|---------------|-----------|----------------|
| 2 | TIR | ISAR | 10.55 | – | 9.6–11.5 $\mu$m | 25° |
| 1 | PMW | EMIRAD-C | – | 7.05 | 7.0365–7.0635 GHz | 55° |
| 1 | PMW | EMIRAD-X | – | 10.69 | 10.59–10.79 GHz | 55° |

## 2.3  Ancillary Data

In this study, a range of datasets that serve as references and support the analyses of the TIR-PMW inter-comparison data was used. To obtain a comprehensive view of the SST in the region of interest throughout the duration of the campaign, Sentinel-3
SLSTR SST L2P data was used (Figure 1). The wind components at 10 m and SST during the campaign were obtained from the European Center for Medium-Range Weather Forecasts (ECMWF) ERA5 reanalysis (Hersbach et al., 2020). Additionally, the Danish Meteorological Institute (DMI) HYbrid Coordinate Ocean Model (HYCOM) v9 data was utilised to provide sea salinity information along the transect (Ponsoni et al., 2023) (Figure 2). PMW SST from the AMSR2 level 2 data was obtained from the JAXA's Global Change Observation Mission 1st – Water (GCOM-W1) platform. This PMW data was employed for
comparing the SST retrievals from the EMIRAD.

## 2.4  Measurement Campaign

The study area is the region between Denmark and Iceland. The ship's track during the measurements (approximately 4853 km), including a stopover in the Faroe Islands, captures the inflow of Atlantic waters into the Nordic Sea and the Arctic, a crucial area associated with the Atlantic Meridional Overturning Circulation (Dickson et al., 2008). Figure 1 illustrates the
ship's trajectory as a black line, while the background image shows the weekly-averaged SST derived from Sentinel 3 Sea and Land Surface Temperature Radiometer (SLSTR) data. The incorporation of SST data from Sentinel 3 SLSTR serves as a reference of the SST conditions during the study period.

The inter-comparison campaign was conducted over a period of 7 days, from May 29 to June 4, 2021. The DMI and the DTU jointly deployed two thermal infrared instruments (ISAR-8 and ISAR-19) and two passive microwave instruments (EMIRAD-
C and EMIRAD-X) onboard the Smyril Line passenger ferry, Norröna, which travels between Denmark and Iceland. The route of Norröna includes stops at the ports of Hirtshals (DK), Tórshavn (FO), Seyðisfjörður (IS), Tórshavn (FO), and Hirtshals (DK) (Figure 1).



The 7-day composite SST indicated warmer waters during the first and last parts of the campaign, from DK–FO and back, ranging between 12 and 16 °C. During the FO–IS (and back) part of the campaign, a sharp SST gradient was crossed where SST dropped from around 8 °C to less than 5 °C.

Throughout the course of the campaign, the weather conditions varied from clear skies to heavy rain. The journey began with clear sky conditions after departure, followed by the development of clouds and the occurrence of mild rain as Norröna approached the Faroe Islands. Subsequently, the sky became partially covered, with a heavy rain event taking place on June 1st as the ferry approached Iceland. For the remainder of the campaign, the sky was partially covered, ranging from 20% to 70% cloud coverage. Additionally, there were instances of fog in the morning and afternoon during the return journey from the Faroe Islands to Denmark (FO-DK). Throughout the duration of the campaign, the sea remained relatively calm, characterized by a low sea state and mild surface roughness conditions. The ISAR recorded the roll, pitch, and azimuth of the instruments (and ship). The mean roll angle recorded was 0.42 degrees, with the highest value of 5.79 degrees observed during the transect between FO-IS. Figure 2 provides additional information on the weather and ocean conditions.

The equipment configuration for the inter-comparison campaign is illustrated in Figure 3. The setup consists of the two ISARs (left), and the two EMIRADs (right) mounted at an approximate elevation of 20 meters above sea level (a.s.l.), above the bridge on the port side of the ship. This configuration was chosen to ensure the observation of undisturbed waters.

## 2.5 TIR-PMW shipborne data

Throughout the campaign, there were minimal instances of precipitation, allowing for almost uninterrupted data collection of SST by the ISAR instruments (ISAR-8 and ISAR-19) at an average sampling rate of approximately 6.9 samples per hour. Regrettably, ISAR-8, being an older generation instrument, experienced a mechanical failure during the initial section (from DK-FO), resulting in the discarding of its data. Thus, only the data collected using ISAR-19 are presented here (top panel in Figure 4).

To ensure truly FRM with observations traceable to SI standards, the SHIPS4SST project developed specific protocols for this shipborne campaign (Høyer et al., 2019). This included pre- and post-calibration against a blackbody reference (CASOTS) (Donlon et al., 2014a). The calibration of ISAR-19 resulted in a mean performance of -0.01 K and a standard deviation of 0.01 K for both the pre- and post-deployment calibrations.

The middle and lower panels of Figure 4 display the measured brightness temperature acquired from the PMW instruments during the field campaign. Intermittent sky measurements were performed throughout the campaign by manually adjusting the antenna orientation, resulting in data points reflecting lower temperatures. The "outliers" at the edge of some of the sky shots are caused by "mixed observations", when data were collected during the motion of the antenna, resulting in a mix of brightness temperature from the sky and the sea surface. An extended period of sky measurements was captured while the ship was anchored at Tórshavn port on the return. The complementing data of the sky serves as a reference for the variability of brightness temperature with minimal geophysical influences (section 4.2), as the sensitivity to the atmosphere in the C and X bands is small (Njoku, 1982).



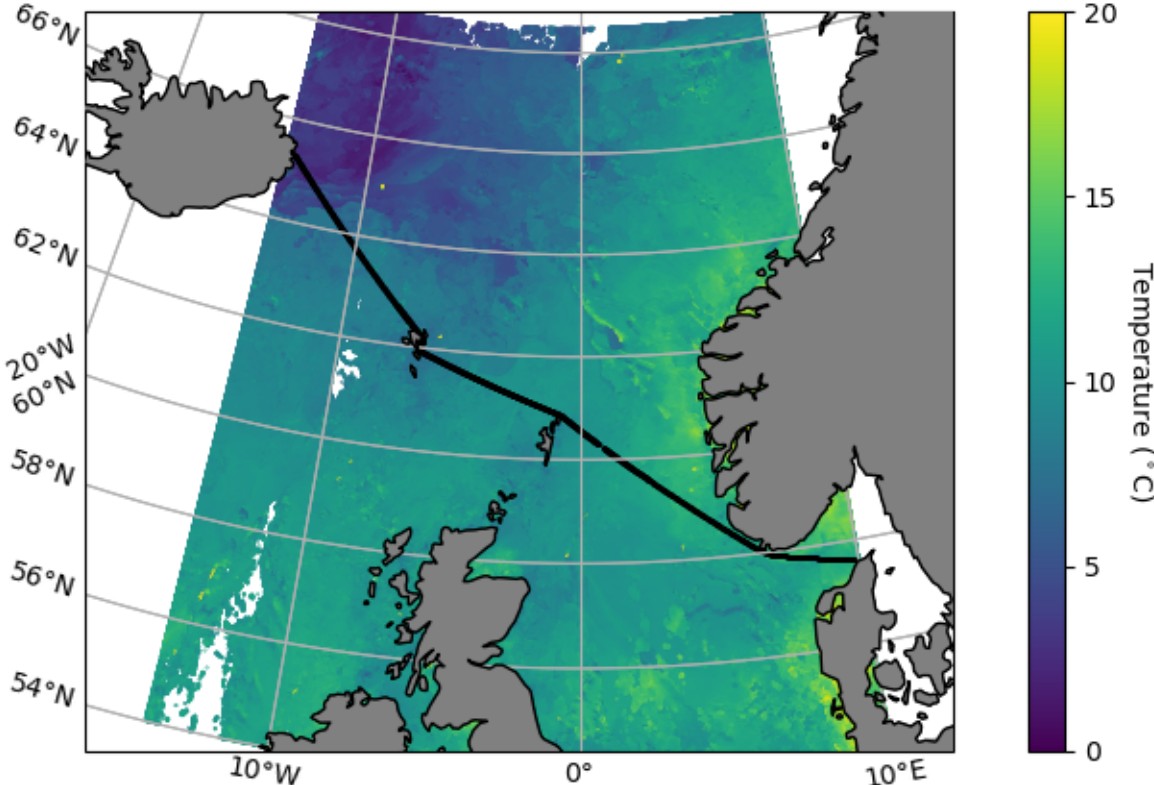

**Figure 1.** Study area. Measurements were made both ways between Denmark and Iceland with stop over in the Faroe Islands. The black line depicts the track position of the ship. The background is the week-averaged SST, from Sentinel 3 SLSTR.

It is important to mention that the C band H-polarization channel (orange dots in the middle panel of Figure 4) showed a persistent noise pattern throughout most of the observational period, which is consistent with previous observations from the static measurements conducted in Copenhagen (Høyer et al., 2021).

The PMW instruments had an average sampling rate of 32 samples per hour for the C band V-pol channel, 16.2 samples per hour for X band V-pol, and 16.8 samples per hour for X band H-pol.



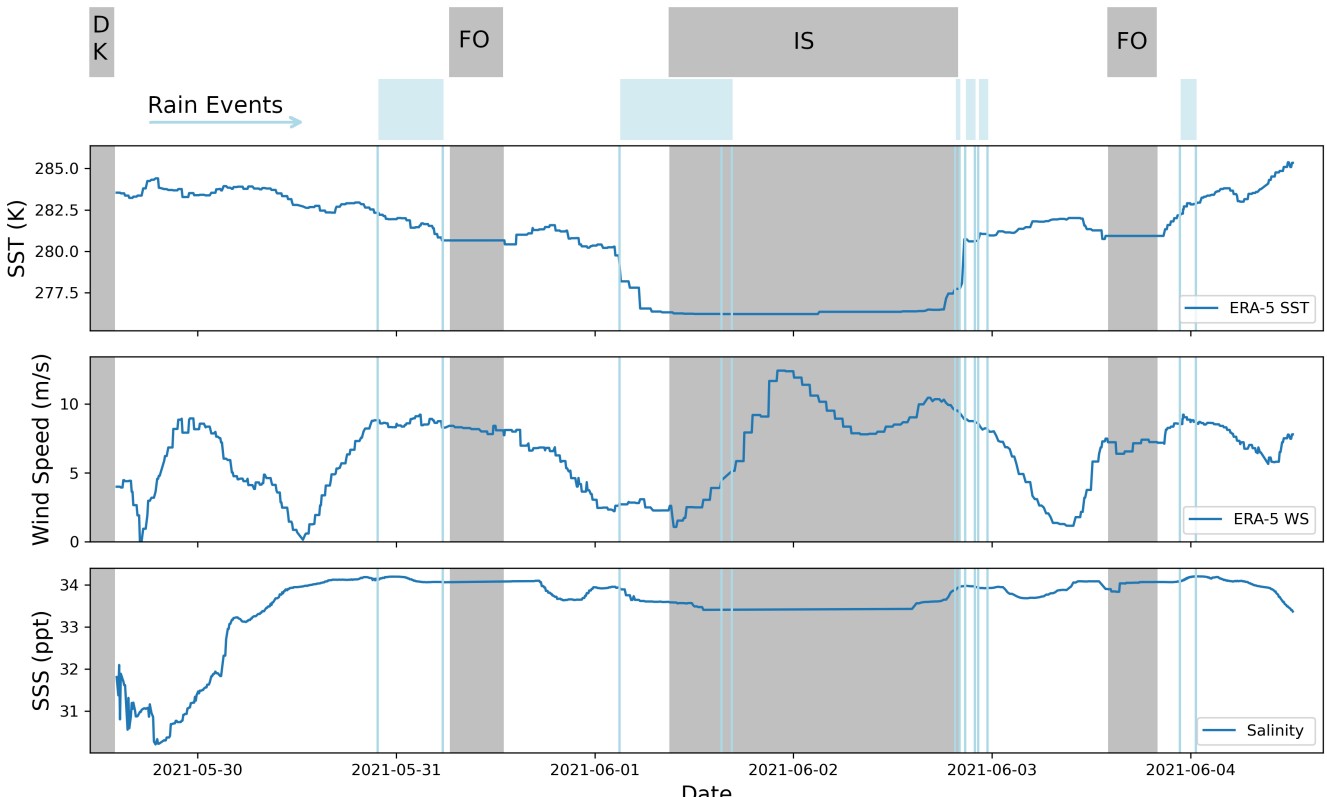

**Figure 2.** Weather and ocean conditions during the inter-comparison campaign. SST and wind speed (WS) are obtained from ERA5 re-analyses data and sea surface salinity from DMI HYCOM model. Grey bands depict the mooring time in the following sequence: Hirtshals (DK) - Tórshavn (FO) - Seyðisfjörður (IS) - Tórshavn (FO). Detected rain event periods are represented by light blue vertical lines.

## 3 Data Processing

### 3.1 Filtering of data

Data obtained from the horizontally polarized channel of the C band (Figure 4) were excluded from the analysis due to the presence of a noisy signal that could not be attributed to geophysical factors. The source of noise can be attributed to
interference originating from RFI although mechanical issues with the cable connection can not be ruled out.

The remaining three channels underwent a filtering process to separate sky measurements and eliminate outliers resulting from instrument manipulation. Special attention was given to the X band H-pol observations, which exhibited consistent systematic offsets between sky measurements. The magnitude of the offsets varied up to a maximum brightness temperature of 15.18 K after a sky measurement on May 30 (Figure 4). The most plausible explanation for these offsets is attributed to
small changes in cable loss caused by mechanical tension in the independent wiring of each channel. This tension arose from the manual movement of the antennas (rotated 90 degrees) to perform sky measurements. To address this issue, the observed



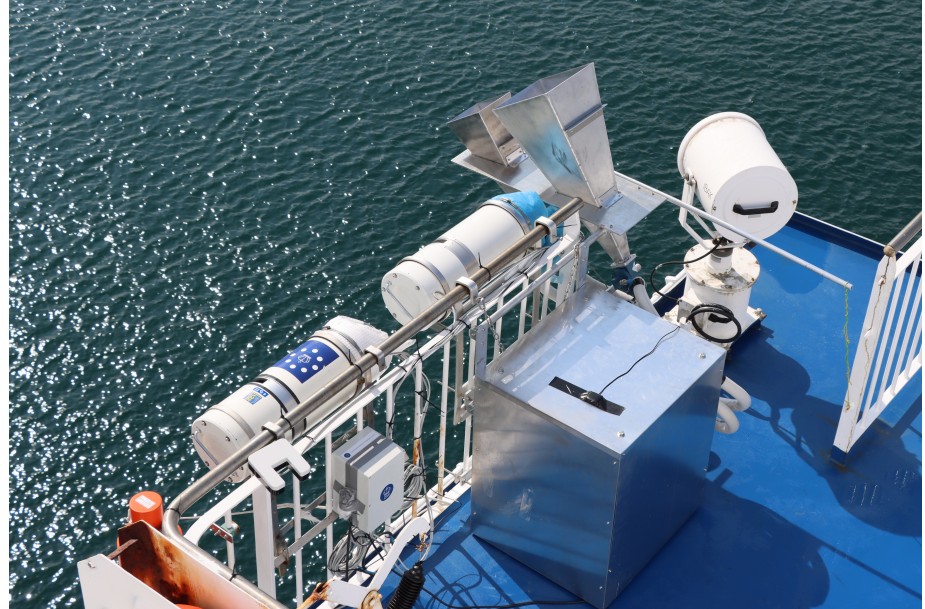

**Figure 3.** Radiometers installed onboard the vessel Norröna. EMIRAD antennas (right side) pointing upwards for performing intermittent sky measurements.

"jumps" in the X band H-pol data were rectified by subtracting the offset from the median within a range of 10 samples before and after each sky measurement. The cumulative sum of these offsets over the entire period amounted to 0.3 K, which supports the notion that these jumps were induced and suppressed by the sky measurements. This adjustment ensured the data integrity
and enhanced the reliability of subsequent analyses.

During the data collection period in Tórshavn, all the sea data obtained by the radiometers had to be excluded from the analysis. This was required as the ship moored with the radiometers directed towards the side road of the peer, rendering the sea measurements invalid.

Data collected with the antennas oriented to the sky was then separated and the sea-oriented dataset was divided into two
categories, i.e. 'moving' data and 'port' data. Subsequently, each analysis was conducted separately, ensuring a thorough examination of these two conditions.

## 3.2 Matchup dataset

The dataset construction process involved matching the EMIRAD dataset, including C band V-pol, X band H-pol, and X band V-pol data within a time window of 300 seconds. Subsequently, the TIR (ISAR-19) data was matched and the obtained dataset
was temporally and spatially aligned with wind components and SST information from ERA5. The maximum allowed time range was 2 hours and the maximum distance range was 0.3 degrees. Additionally, the dataset was aligned with the salinity



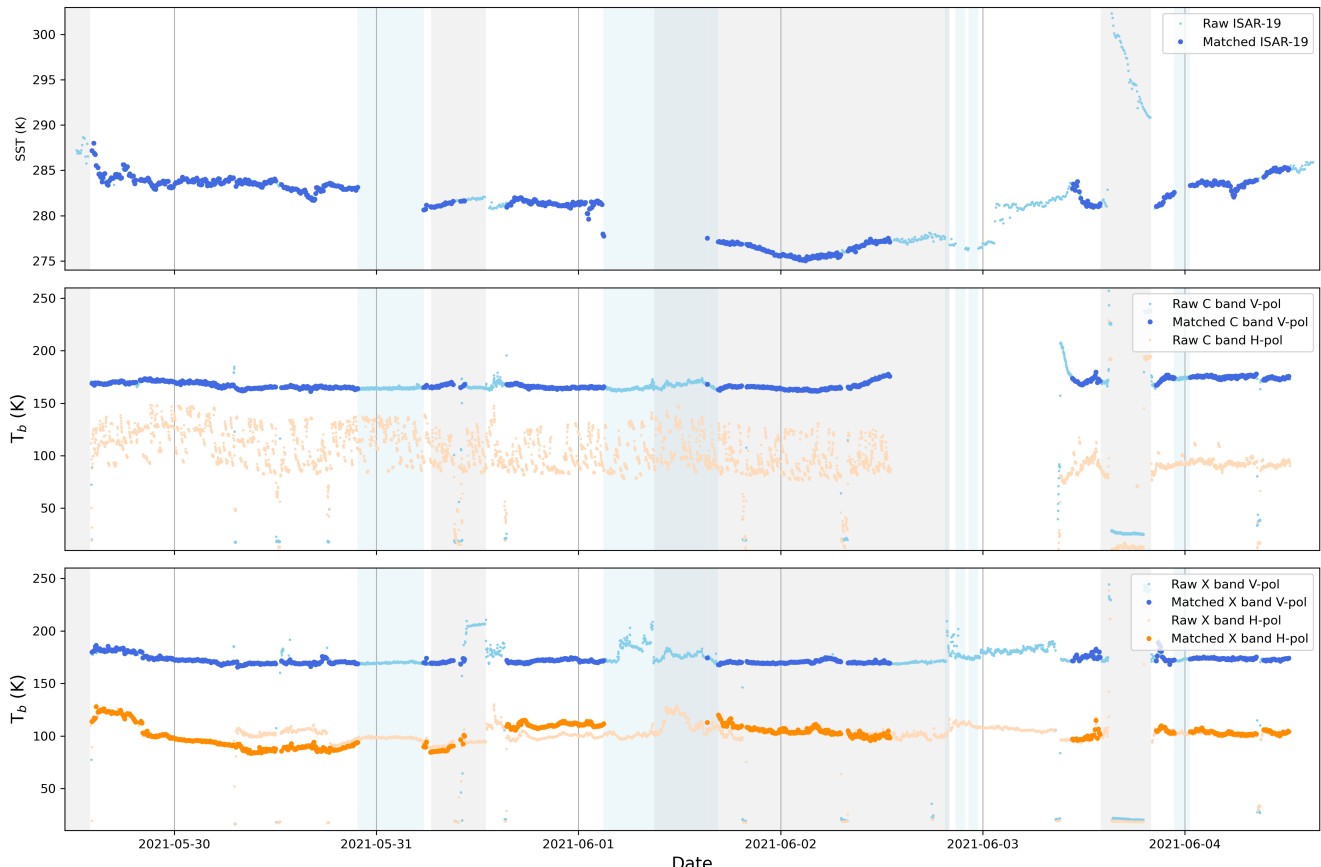

**Figure 4.** Original and matched data from the TIR and PMW instruments. SST from ISAR-19 (top), brightness temperature measurements from C (middle), and X band (bottom). Light colors indicate raw data, dark colors depict the resulting match-up dataset of observations. Vertical shaded bands indicate port time (grey) and rain events (blue), described in Figure 2.

output of the DMI HYCOM forecasting model. This process resulted in a dataset of 708 points (N) which are depicted in dark colors in Figure 4 and are further used in the SST retrieval algorithm.

## 4 Microwave brightness temperature ($T_b$) characteristics

### 4.1 Instrumental noise

The instrumental noise was assessed from sky measurements, which provide information on the stability of the instrument when there is minimal geophysical effect.

The measurements were conducted for a duration of 4 hours at Tórshavn port on June 3rd, with the antennas oriented upward. Throughout this period, the sky conditions exhibited intermittent presence of thin clouds, covering approximately 20-40% of




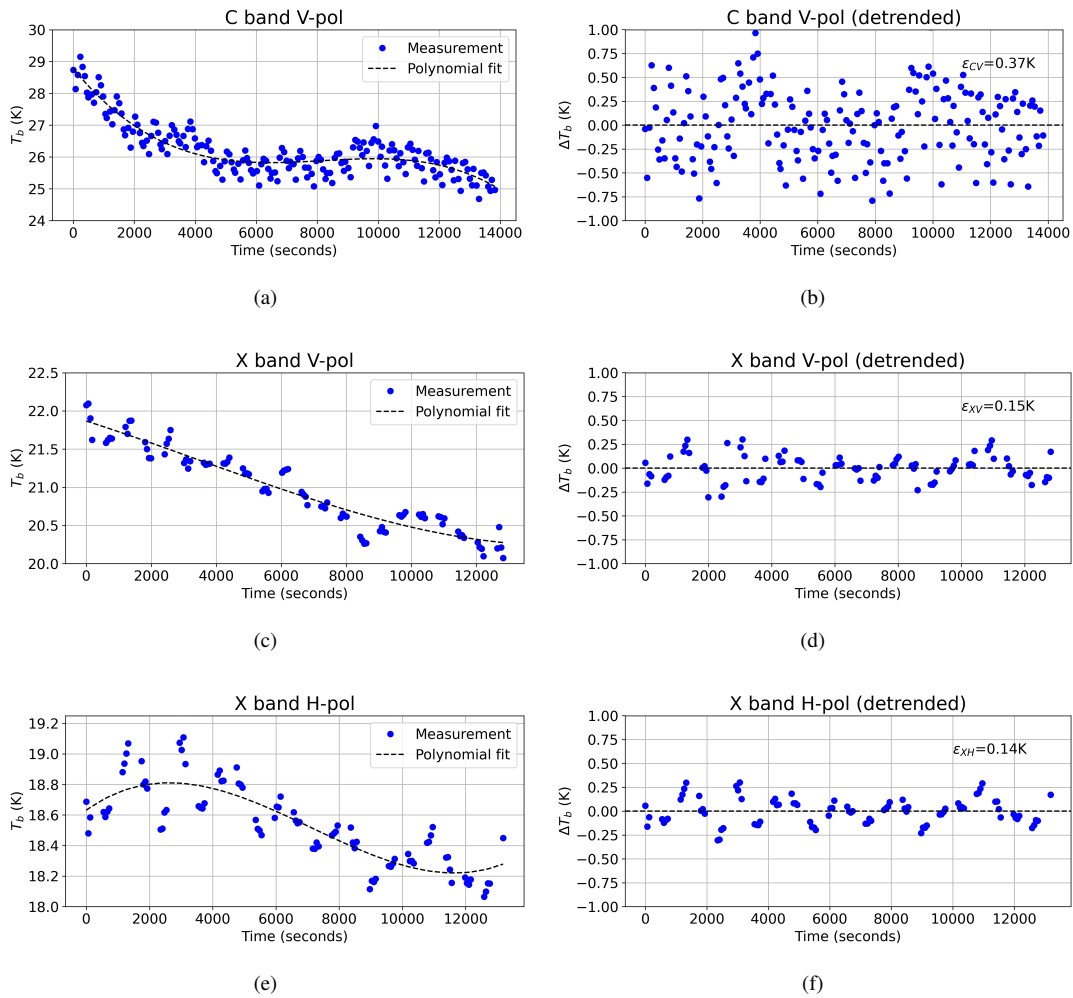

**Figure 5.** Time series of sky measurements at Tórshavn (a, c, e) and the corresponding de-trended signal (b, d, f) using a 3rd-degree polynomial fit. The instrumental error ($\epsilon_{inst}$) is the standard deviation of the residuals.

the sky. This particular set of sky measurements was employed to assess the stability of the instruments, as it represents the longest continuous sky observation conducted during the campaign. Figure 5 depicts the $T_b$ variability of sky observations from the EMIRAD instruments.

The instrumental error (random noise) was quantified from the $T_b$ variability of the sky measurements shown in Figures 5a, c and e, which consist of both the geophysical variability (changes in the sky condition) and random noise. A cyclic pattern can be

noticed in the X-band sky variability plots (around half an hour period), which is likely the result of temperature regulation that produced slow changes of the signal. Assuming that the sky condition varies slower than the noise, a de-trending process was applied to the time series of the sky measurements by subtracting a polynomial fit from the original time series. Subsequently,



the standard deviation of the residuals (de-trended signal, see Figure 5b, d and f) was calculated and used as an estimate of the random instrumental error. The appropriate order of a polynomial used for a fit was determined through a sensitivity test; the

standard deviation of the residuals reached stability from the 3rd-degree polynomial, thus it was selected for the de-trending process. The instrumental errors, i.e. standard deviation of the residuals, for C band V-pol, X band V-pol, and X band H-pol were determined to be 0.37 K, 0.15 K, and 0.14 K, respectively.

## 4.2 Observed $T_b$ variability

The variability of $T_b$ data was evaluated individually for each channel using the raw clean dataset (filtered outliers and sky
measurements). This assessment was performed by measuring the standard deviation of the absolute differences between each data point and the mean value within a specific time or space window. Figure 6a shows the standard deviation of $T_b$ for each channel at intervals from 5 minutes to 60 minutes for the moving data and TIR SST is included for reference. In all cases, there was a steeper increase of the standard deviation from 5 to 20 minutes, particularly obvious for the X band H-pol, which also shows the highest values. The V-pol for both C and X bands (blue and green dots) indicates similar temporal variability
increasing from 0.6 at 5 minutes to approximately 0.8 at 20 minutes, beyond which a slow increase up to 1.07 and 1.11 respectively, at 60 minutes occurred. The ISAR SST standard deviation was below 0.1 at 5 minutes and slowly increased up to 0.38 at 60 minutes. When port time was considered, Figure 6c indicates a higher temporal variability for the passive microwave channels, especially for the X band H-pol, although the ISAR SST remains stable at 0.12 K.

    The spatial variability assessed for distances of 5 km up to 50 km is shown in Figure 6b, where a similar pattern to the
temporal variability is identified although standard deviation values are overall slightly higher for all instruments and channels.



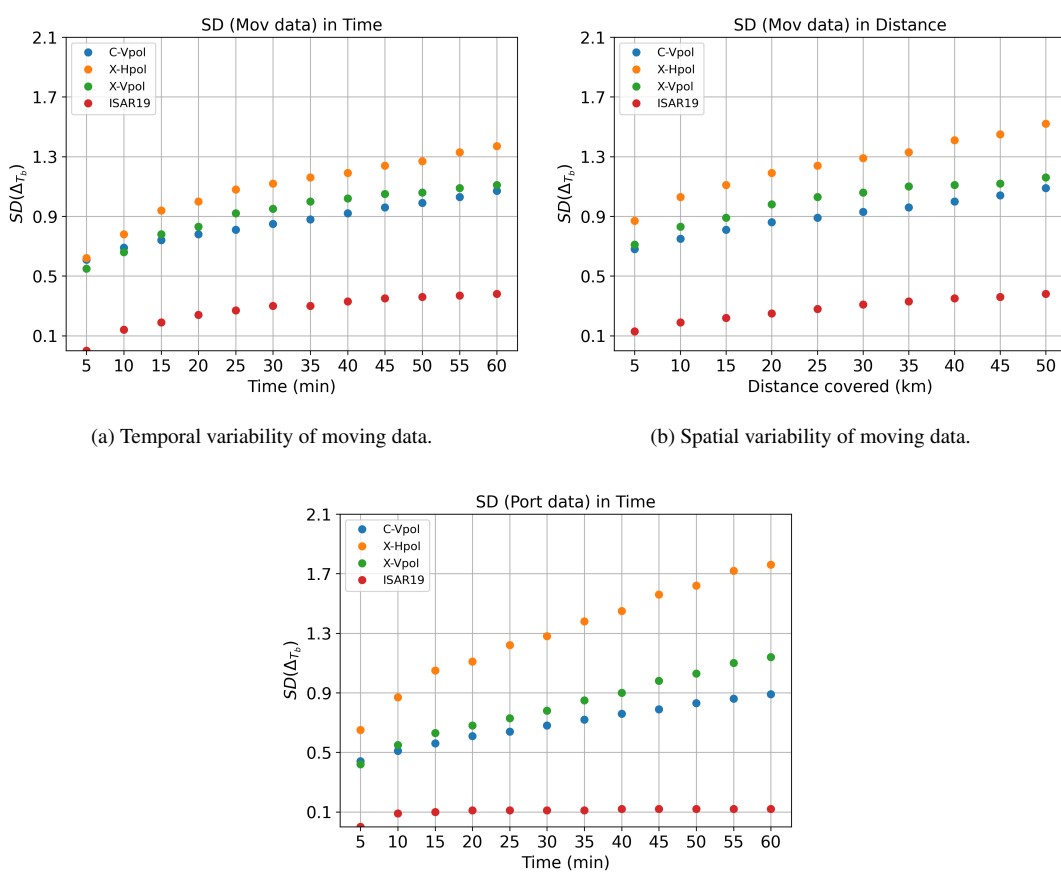

(a) Temporal variability of moving data.

(b) Spatial variability of moving data.

(c) Temporal variability of port data.

**Figure 6.** Variability of data collected by all instruments, measured as the standard deviation of data collected in relation to time and distance, for both moving and port data.

## 4.3 Sensitivity of $T_b$ to geophysical parameters based on simulations

To investigate the sensitivity of microwave $T_b$ to various geophysical parameters, a microwave forward model was employed, following the methodology described in Wentz (2002); Nielsen-Englyst et al. (2021). It is important to note that the forward model employs slightly different frequencies (6.925 GHz and 10.65 GHz for C and X bands, respectively) compared to EMI-
RAD (Table 1).

The input parameters for the forward model are SST, sea surface salinity (SSS), wind speed (WS), incident angle ($\theta$), the angle between the azimuth of the ship and the wind direction (relative angle, $\phi_r$), total column water vapor (TCWV), and total column liquid water (TCLW). As the measurements were taken near the surface within the C and X bands, the two parameters related to atmospheric effects (TCWV and TCLW) were set to zero throughout this study, assuming negligible atmospheric
impact on the measured $T_b$ (Njoku, 1982).



The analysis focused on examining how the microwave $T_b$ changes in response to variations in the input parameters for the forward model. The reference values used for the test were: SST = 280 K, WS = 5 m s$^{-1}$, SSS = 35‰, $\theta = 55°$, $\phi_r = 180°$.

The results are shown in Figure 7, where the symbol $\Delta$ indicates the deviation from a reference value. Large changes in $T_b$ were induced by changes in WS (especially for the X band H-pol channel), $\theta$ for all channels and SST (especially for the C band V-pol channel). The contributions of salinity and relative angle were small for all channels.

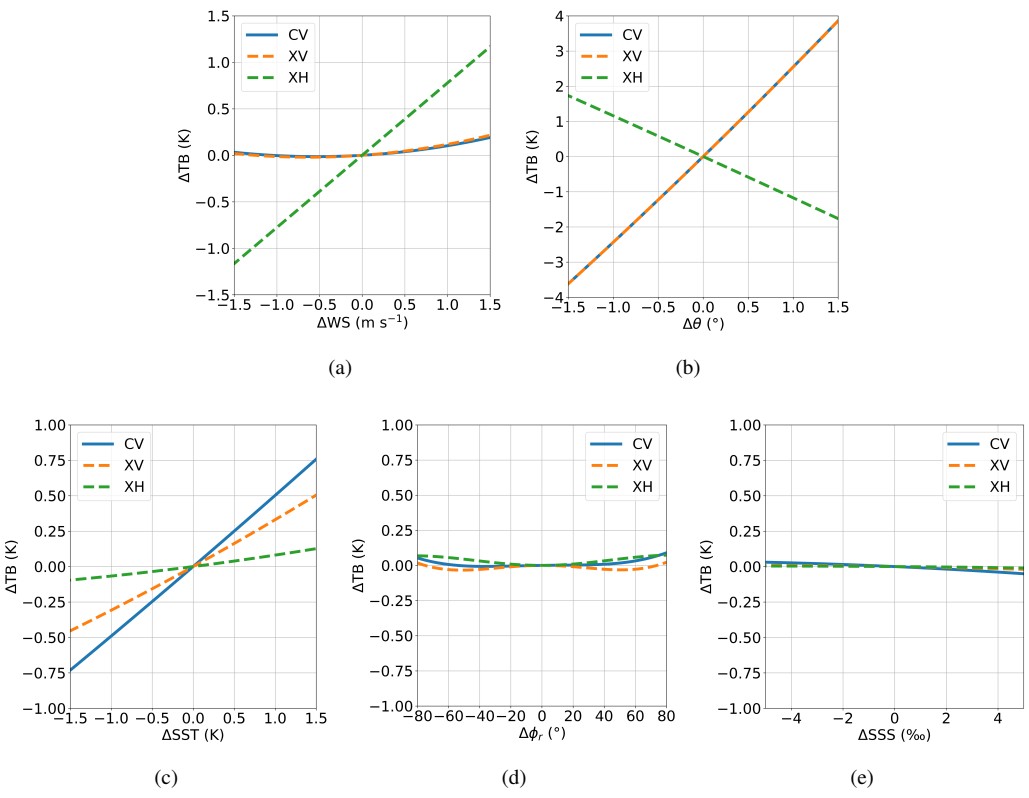

**Figure 7.** Brightness temperature change of C and X bands simulated by the forward model for a) WS, b) $\theta$, c) SST, d) $\phi_r$ and e) SSS.

## 5 PMW SST

### 5.1 Regression analysis

The retrieval method used to derive SST from PMW $T_b$ measurements in this study was based on Alerskans et al. (2020). To optimize the linear regression algorithm, multiple iterations were conducted, considering input parameters and statistical outputs of the fit. Initially, the incident angle $\theta$ and the relative angle $\phi_r$ were also included. However, the incident angle derived from the ISAR-19 sampling was not representative of the instant incident angle of the matching PMW data points. As a result, only the standard deviation of the ship's roll between samples was used as a measure of the incident angle error. The



sensitivity analysis of $\phi_r$ indicated its low impact on $T_b$, leading to its exclusion from the retrievals equation. Conversely, WS was included as a predictor due to its significant influence on $T_b$, as demonstrated in the previous section. The final equation used for the regression analysis is as follows:

$$SST_{MW} = c_0 + c_1 t_{CV} + c_2 t_{CV}^2 + c_3 t_{XV} + c_4 t_{XV}^2 + c_5 t_{XH} + c_6 t_{XH}^2 + c_7 WS + c_8 WS^2 + 1/\epsilon \tag{1}$$

The variable $t$ represents $T_b - 150$ and the subscripts of $t$ denote the specific PMW band and polarization involved. The term $\epsilon$ represents the observational error associated to the instruments and the input parameters, as shown in Equation 2, where the subscript $p$ refers to the parameters inducing errors. These are instrumental, WS, and incident angle errors. The accuracy of $SST_{IR}$, $\epsilon_{IR}$, was determined to be 0.01 K from the pre- and post-deployment calibration process. For the PMW instruments, the estimated instrumental error from sky measurements (as depicted in Figure 5) was used. The WS error ($\epsilon_{WS}$) was assumed to be 2 m/s (Nielsen-Englyst et al., 2018). Furthermore, the standard deviation of the ship's roll ($\epsilon_\theta$) recorded by ISAR-19 for each sample was used as a reference of the incident angle error.

$$\epsilon = \sqrt{\sum_p \epsilon_p{}^2} \tag{2}$$

The regression coefficients in Equation 1 were calculated using $SST_{IR}$ from ISAR-19 as the independent variable. These coefficients were computed based on a randomly selected "training" dataset, which comprised two-thirds of the matchup data. Equations 1 and 2 were separately applied to the three training datasets: all data, moving, and port, in order to observe the output under distinct conditions. Considering the minimal roll during the mooring period and the limitations of ERA5 data near land, the wind speed was set to zero for the two port periods under consideration. The resulting coefficients obtained from this analysis are presented in Table 2. The remaining matchup dataset ("test") was used for retrieving the sea surface temperature ($SST_{MW}$) and for further analysis.

**Table 2.** Coefficients resulting from the regression equation applied to datasets separately.

| $c$ | All | Moving | Port |
|---|---|---|---|
| $c_0$ | 284.43 | 285.009 | 302.942 |
| $c_1$ | 0.804 | 0.832 | 0.703 |
| $c_2$ | -0.014 | -0.015 | -0.015 |
| $c_3$ | 0.085 | -0.350 | -1.026 |
| $c_4$ | -0.001 | 0.007 | 0.023 |
| $c_5$ | 0.814 | 0.379 | 1.048 |
| $c_6$ | 0.009 | 0.004 | 0.011 |
| $c_7$ | 1.688 | 0.081 | 0 |
| $c_8$ | -0.139 | -0.017 | 0 |





## 5.2 Uncertainty estimation

An uncertainty propagation was performed in order to identify the main uncertainty components and the expected total retrieval
uncertainty of the retrieved PMW SST. The uncertainty resulting from a certain parameter is quantified as the standard deviation

of the retrieved SST distribution when subjected to perturbations in that parameter. This analysis utilized the microwave forward
model described in Section 4.3. Taking into account the possibility of a systematic bias between the forward model and actual
observations, our focus is solely on measuring the variation in retrieved SST induced by specific perturbed parameters.

**Table 3.** Reference values for input parameters in the forward model, with the uncertainty perturbation denoted within parentheses.

| Parameters (unit) | Reference values (uncertainty) | |
| --- | --- | --- |
| | Moving | Port |
| Sea surface temperature (K) | 281.35 | |
| Salinity (‰) | 33.3 (1.18) | |
| Relative Angle (°) | 245.2 (81.85) | 0 (0) |
| Wind speed (m s$^{-1}$) | 6.25 (2) | 0 (0) |
| Incident angle (°) | 55 (0.2) | 55 (0) |
| TB C band V-pol (K) | 159.49 (0.37) | 160.58 (0.37) |
| TB X band V-pol (K) | 163.70 (0.15) | 164.98 (0.15) |
| TB X band H-pol (K) | 80.13 (0.14) | 75.22 (0.14) |

To evaluate the components in the uncertainty budget and estimate the total uncertainty, the first step involved setting ref-
erence values for the input parameters in the forward model (Table 3) for moving and port cases. The parameters examined

include SST, salinity (SSS), relative angle ($\phi_r$), wind speed (WS), and incident angle ($\theta$). The reference values were derived by
averaging the corresponding data points for moving and port data. Subsequently, by inputting the described reference values
into the forward model, the reference T$_b$ for the three channels were obtained.

Predictors in Equation 1 are here referred to as explicit parameters (i.e., T$_b$ and WS) and their uncertainties were defined in
the previous sections. The parameters not used as a predictor, hereafter referred to as implicit parameter, including SSS, $\theta$, and

$\phi_r$, their uncertainty was obtained as the variability ($\sigma$) during the observation period. The variability caused by the roll on the
incident angle ($\epsilon_\theta$) was 0.2°, which is the average standard deviation of roll angle measurements obtained by the ISAR-19 for
moving data, whereas for the port data, $\epsilon_\theta$ was close to zero.

In the next step, a total of 100,000 samples for the parameter of interest were generated randomly. The samples followed a
Gaussian distribution with a mean value equal to the parameter's reference value, and a standard deviation of its uncertainty. For

the subsequent step, distinct calculations were performed for the implicit and explicit parameters. For the implicit parameters,
the generated distribution of a target parameter was inputted into the forward model along with the reference values of the
remaining parameters. This process resulted in the generation of distributions of T$_b$ for each channel (i.e. C band V-pol, X
band V-pol, and X band H-pol). These T$_b$ distributions were then incorporated into the regression equation (Equation 1) with





coefficients derived from the entire dataset, resulting in a distribution of SST. This analysis enables us to evaluate the level

of uncertainty that arises from excluding the implicit parameters in the retrieval process, which are varying and affecting the

microwave $T_b$. As for the explicit parameter, the generated distribution was directly used in the regression equation to obtain

the SST distribution. Finally, the corresponding standard deviation values of the resulting SST distributions were calculated.

    Additionally, an error propagation analysis was performed incorporating the instrumental error. 100,000 random $T_b$ samples

were generated for the three channels, following a Gaussian distribution. These samples had means corresponding to their refer-

ence values and standard deviations representing the instrumental error. Unlike the previous case, involving implicit parameters,

where the $T_b$ distributions were correlated, in this scenario, the distributions for each channel were independent. Subsequently,

the $T_b$ distributions were utilised in the regression equation to estimate the standard deviation of the SST distribution.

    Once the uncertainties associated with individual parameters (contributors) were obtained, the total uncertainty SST in-

duced by these parameters was calculated with the Equation 2. This calculation assumes that the uncertainty contributors are

independent.

**Table 4.** Uncertainty contributions to SST retrieval with induced values for each channel for moving and port conditions.

| Contributor | Moving | | | | Port | | | |
|---|---|---|---|---|---|---|---|---|
| | $\epsilon_{CV}$ | $\epsilon_{XV}$ | $\epsilon_{XH}$ | $\epsilon_{SST}$ (K) | $\epsilon_{CV}$ | $\epsilon_{XV}$ | $\epsilon_{XH}$ | $\epsilon_{SST}$ (K) |
| $\epsilon_{inst}$ | 0.37 | 0.15 | 0.14 | 0.21 | 0.37 | 0.15 | 0.14 | 0.20 |
| $\epsilon_{WS}$ | 0.44[a] | 0.48[a] | 1.71[a] | 0.80 | - | - | - | - |
| $\epsilon_{SSS}$ | 0.01 | 0.00 | 0.00 | 0.00 | 0.01 | 0.00 | 0.00 | 0.00 |
| $\epsilon_{\theta}$ | 0.50 | 0.50 | 0.23 | 0.39 | - | - | - | - |
| $\epsilon_{\phi_r}$ | 0.21 | 0.26 | 0.05 | 0.13 | - | - | - | - |
| Skin-subskin RMSE | - | - | - | 0.28[b] | - | - | - | 0.28[b] |
| Total uncertainty | 0.79 | 0.76 | 1.73 | 0.97 | 0.37 | 0.15 | 0.14 | 0.34 |

[a] These values were not used to calculate $\epsilon_{SST}$

[b] (Wurl et al., 2019)

    The uncertainty derived from the perturbed input parameters analysis are summarized in Table 4. The instrumental error

contributed to an uncertainty of approximately 0.2 K, denoted as $\epsilon_{SST}$. The uncertainty in salinity, represented by $\epsilon_{SSS}$, had a

negligible effect on $T_b$, and therefore, had minimal influence on the overall $\epsilon_{SST}$ for both conditions.

    In the case of moving data, the uncertainty in wind speed (surface roughness), denoted as $\epsilon_{WS}$, had the greatest impact on

$\epsilon_{SST}$ among the contributing factors, resulting in an uncertainty of 0.8 K. When considering the uncertainty resulting from the

incident angle due to the ship's roll, it was estimated that an uncertainty of 0.2° in $\epsilon_{\theta}$ leads to 0.5 K uncertainty in vertically

polarized $T_b$ and 0.23 K in horizontally polarized $T_b$. These uncertainties contribute to an overall of 0.39 K in $\epsilon_{SST}$. On the

other hand, the effect of 81.85° variation in $\phi_r$ had a minimal influence, inducing only 0.13 K uncertainty when propagated

through the retrieval equation. This supports the previous decision to exclude $\phi_r$ from the retrieval process.





Moreover, it is important to account for the variability between skin and sub-skin SSTs in the uncertainty estimation. In situ measurements by Wurl et al. (2019) reveal a strong correlation between skin and sub-skin SSTs, with an RMSE of 0.28 K. Although this was obtained from different latitudes, it is here used as a reference of this geophysical component.

Consequently, the estimated total uncertainty of the retrieval of SST was 0.97 K for data collected while moving, whereas the uncertainty for the stationary time was smaller, estimated to be 0.34 K.

## 5.3   Comparisons of PMW and TIR SST

Figure 8 presents scatter plots that depict the relationship between $SST_{MW}$ and $SST_{IR}$, along with the corresponding coefficient of determination ($R^2$) indicating the goodness of the fit. Uncertainty values of the $SST_{MW}$ retrieval have been added to Figure 8b and 8c as analysed in the previous section and demonstrate that the derived uncertainties for the PMW retrievals are sensible. When considering all data (Figure 8a), the obtained $R^2$ value was 0.88, indicating a strong overall correlation between

$SST_{MW}$ and $SST_{IR}$. However, when only moving data were considered (Figure 8b), the $R^2$ decreased to 0.45, indicating a weak correlation between $SST_{MW}$ and $SST_{IR}$. The SST values ranged from 280 to 286 K, with a positive mean difference between $SST_{MW}$ and $SST_{IR}$. In contrast, the port data (Figure 8c) primarily comprised cold water observations (IS), with SST values ranging from 275 to 278 K. In this case, the $R^2$ of 0.83 indicates a better agreement between $SST_{MW}$ and $SST_{IR}$ compared to the moving dataset. Nevertheless, some discrepancies were noted for the data collected in the slightly warmer

waters of Tórshavn (FO).

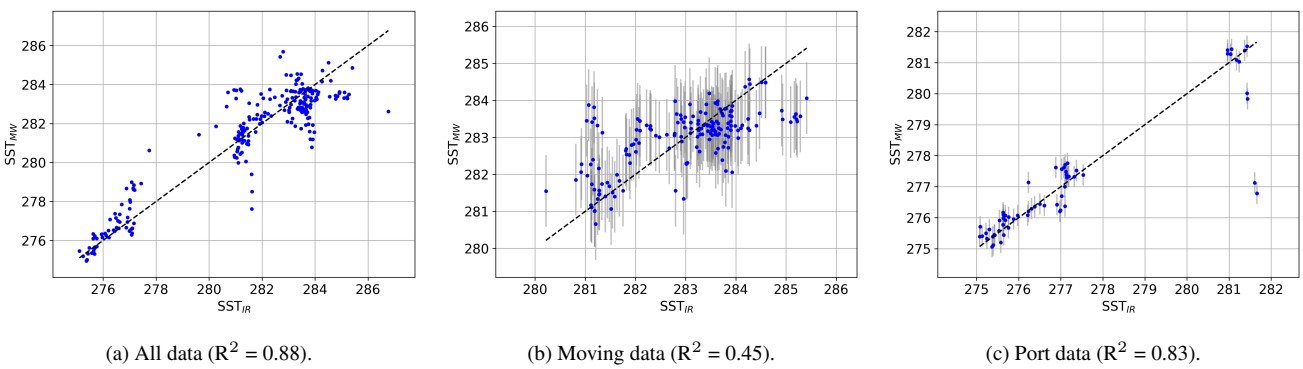

(a) All data ($R^2$ = 0.88).      (b) Moving data ($R^2$ = 0.45).      (c) Port data ($R^2$ = 0.83).

**Figure 8.** Scatter plot comparing $SST_{IR}$ and retrieved $SST_{MW}$ values. a) Retrievals evaluated for the complete dataset, and separately for b) moving data and c) port data. Grey bars depict the uncertainty estimations obtained in Section 5.2

Figure 9 illustrates the time series of input variables and output SST of the retrieval process. The top panel displays the input variable $SST_{IR}$ plotted alongside the retrieved $SST_{MW}$, which is the combined result obtained from both the moving and port data. Especially for the first part of the campaign, before the first rain event (blue shaded area) there is good agreement between the two SSTs which remains the case up to June 2nd when the ship was moored (IS). The agreement during the last part of the

campaign, after June 3rd is deteriorating with the $SST_{IR}$ showing more variability compared to the $SST_{MW}$. The time series




of $T_b$ for the V-pol from both X and C bands are shown in the second panel, $T_b$ for the H-pol from the X band is shown in the third panel while WS and $\epsilon_\theta$ are shown in the bottom panel.

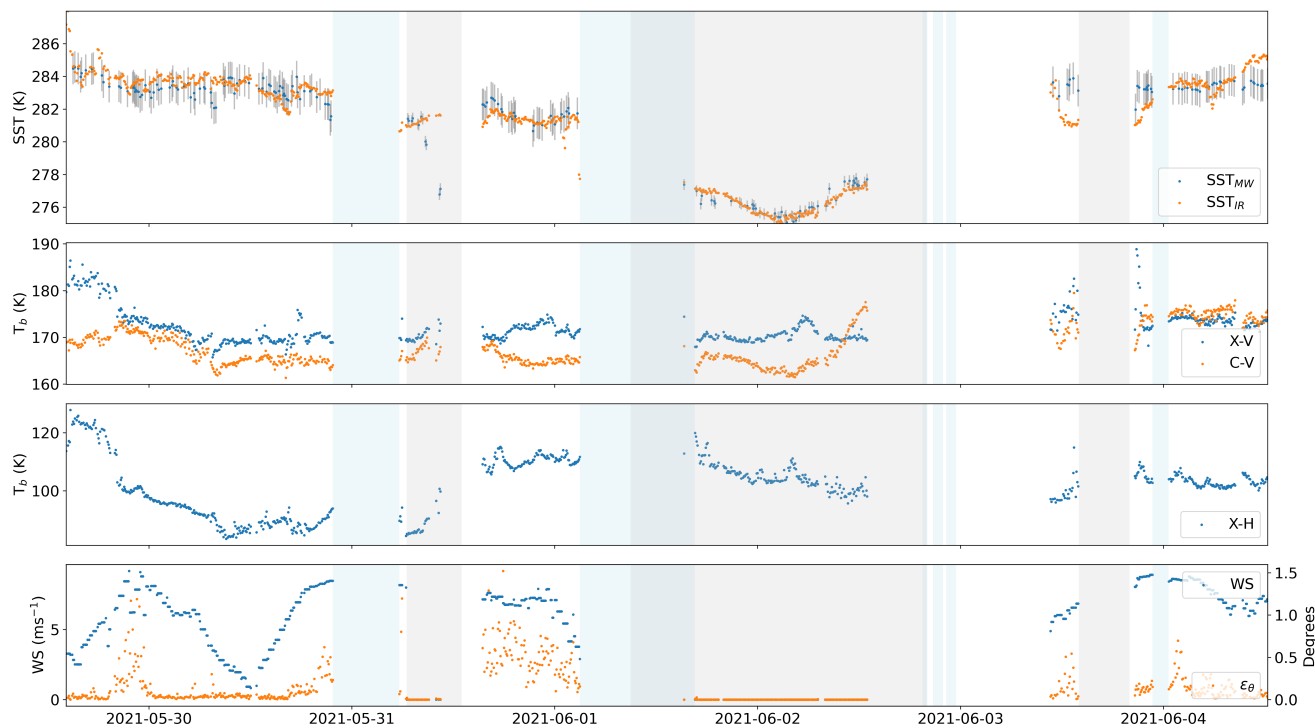

**Figure 9.** Matchup data used in the comparison of SST from ISAR-19 and SST retrieved from EMIRAD throughout the campaign. From top to bottom: $SST_{MW}$ (separately obtained for moving/port data) with error bars and $SST_{IR}$, vertically polarized $T_b$, horizontally polarized $T_b$, WS, and $\epsilon_\theta$.

Table 5 shows the statistics of the comparison between $SST_{MW}$ and $SST_{IR}$. When considering all data the mean difference was –0.06 K, indicating a minimal systematic bias. The RMSE was 1.13 K, reflecting the overall variability between the two signals. During the moving periods, the mean difference was closer to zero at 0.02 K, and the RMSE significantly decreased to 0.88 K. However, during port docking, the mean bias slightly increased to –0.09 K, and the RMSE slightly rose to 0.94 K.

To examine the potential impact of diurnal variability in atmospheric conditions on the sea surface, a comparative analysis of $SST_{MW}$ was conducted, as depicted in Figure 10. The data were segregated into two categories based on the delineation of day and night, with the time boundaries set at 8:00 and 22:00 UTC.

When all data were considered, there was a wider range of differences during day-time (10a) compared to night-time (10d) and although the mean bias $\mu$ was smaller, the standard deviation $\sigma$ was higher. This pattern of higher $\sigma$ and wider distribution of biases for day-time compared to night-time was consistent also for the moving (10b, 10e) and port data (10c, 10f) and can be attributed to diurnal variability of the SST and the difference between skin ($SST_{IR}$) and sub-skin ($SST_{MW}$) temperatures.



**Table 5.** Comparison of SST retrieved from PMW $T_b$ using the regression analysis and TIR observations from ISAR-19.

|         | All    | Moving | Port   |
|---------|--------|--------|--------|
| $\mu$   | -0.06  | 0.02   | -0.09  |
| $\sigma$ | 1.12  | 0.88   | 0.93   |
| RMSE    | 1.13   | 0.88   | 0.94   |
| $R^2$   | 0.88   | 0.45   | 0.83   |
| N       | 234    | 171    | 64     |

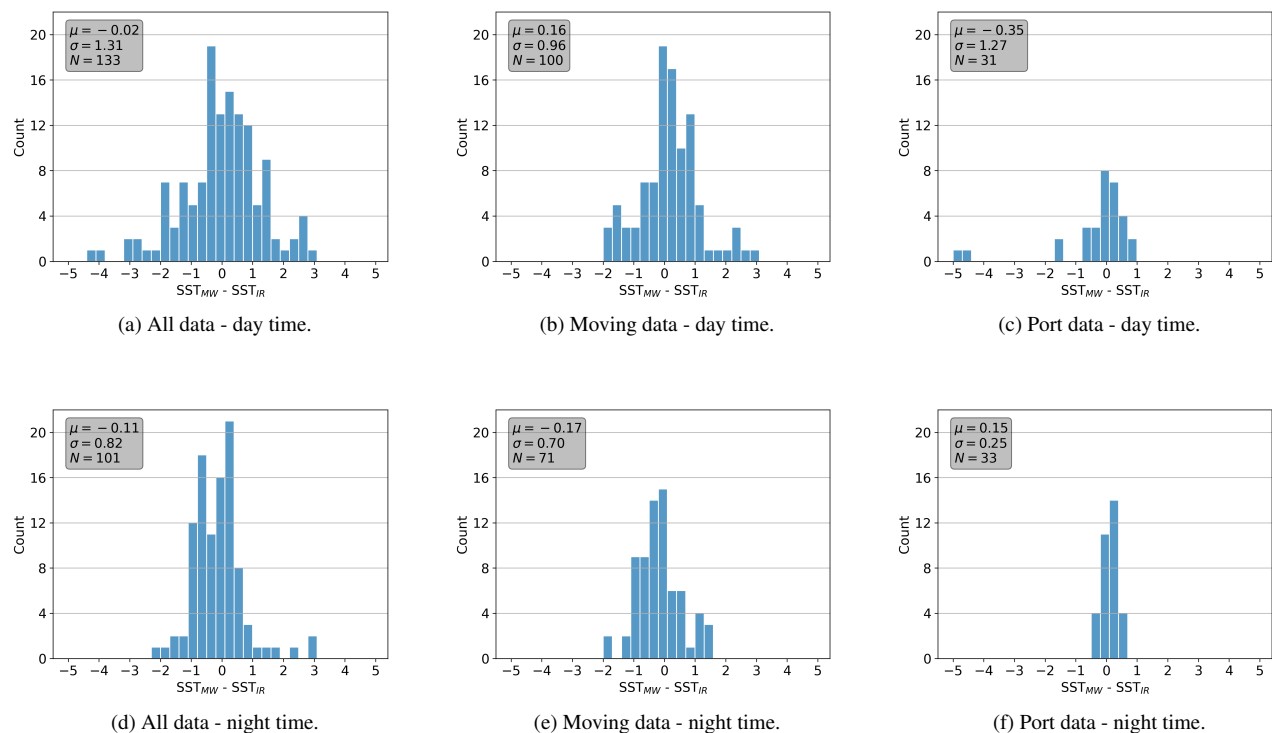

**Figure 10.** Histogram of the difference between the retrieved $SST_{MW}$ and $SST_{IR}$ for day and night conditions.

The analyses of the moving data shows that the mean bias during day-time is positive (0.16 K), while during night-time it is
negative (-0.17). This suggests that there is no significant effect of the diurnal near-surface warm layer on the bias (Gentemann et al., 2003; Gentemann and Minnett, 2008; Alappattu et al., 2017).

## 5.4 Comparison to Satellite products.

To asses the bias of the retrieved SST from EMIRAD against available SST products, data from Sentinel 3 SLSTR and AMSR2 level 2 (10GHz) were utilised. The satellite data were separately matched to the retrieved ("test") data subset by considering a




time window of 3 hours and a spatial window of 0.1 degree. This matching process resulted in 53 SLSTR data points and 40
AMSR2 data points.

     Figure 11 illustrates the scatter plot and histogram of the comparison between EMIRAD's retrieved SST and SLSTR, followed by the comparison to AMSR2.

     SLSTR and EMIRAD SST appear to be in good agreement, with a mean bias of 0.3 K and standard deviation of 0.9

K (Figure 11b), $SST_{skin}$ being colder. On the other hand, when comparing microwave derived $SST_{subskin}$, AMSR2 shows warmer temperatures than those retrieved from EMIRAD ($\mu$ = -0.87 K) and higher variability ($\sigma$ = 1.07 K) (Figure 11d).

     Despite the relatively large temporal and spatial windows used for the search of matching data point, no correlation was found between the magnitude of the bias and the distance or time difference of the compared SST values.

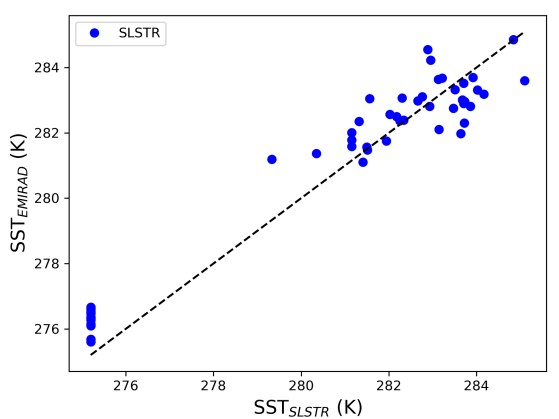
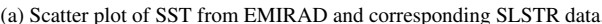

(a) Scatter plot of SST from EMIRAD and corresponding SLSTR data.

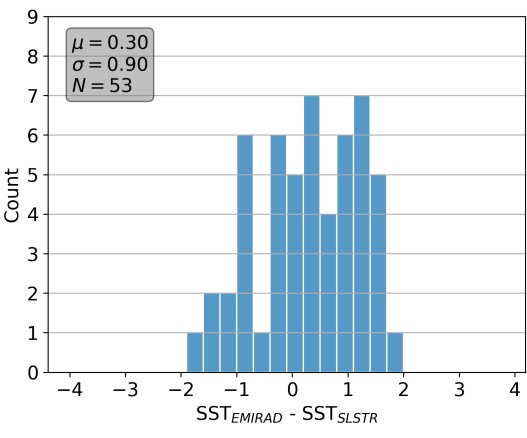

(b) Bias between SLSTR and EMIRAD.

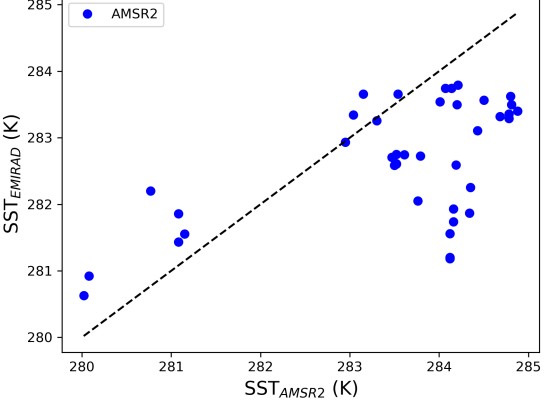
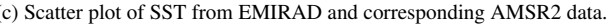

(c) Scatter plot of SST from EMIRAD and corresponding AMSR2 data.

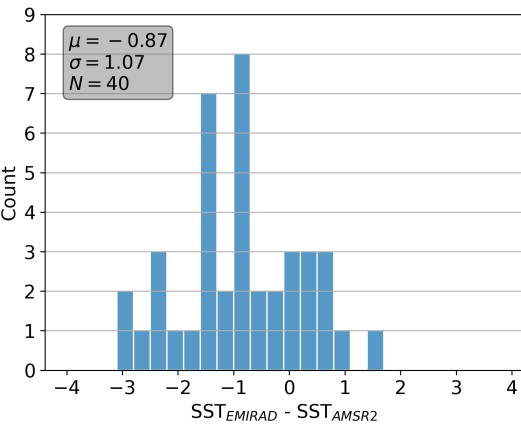

(d) Bias between AMSR2 and EMIRAD.

**Figure 11.** Comparison of EMIRAD retrieved SST against satellite products. Data sets were matched separately.



## 6 Discussion

This study presents a unique comparison of sea surface temperature (SST) obtained from simultaneous thermal infrared (TIR) and passive microwave (PMW) radiometer measurements during a week-long shipborne campaign from Denmark to Iceland in the early summer of 2021.

It is important to note that while the ISAR instrument is a fully automated, stable, and well-documented instrument, widely used as a reference for satellite validation products (Donlon et al., 2014b; Wimmer et al., 2012; Wimmer and Robinson, 2016),
the EMIRAD instrument is more experimental and had been refurbished specifically for this campaign, requiring manual operation at times.

The matchup dataset was constructed by considering a time window of up to 5 minutes between the actual observed values, without performing any sample averaging. The choice of the time window length was based on the lowest sampling rate among the four instruments involved (i.e. ISAR-19). This decision led to a reduced dataset for the comparison, presenting a challenge
for the data analyses and chosen methodologies due to the small number of data points available when separating them into different categories (i.e., moving, port, day, and night). The resulting matchup dataset used for the comparison also excluded the C band H-pol given the noisy signal obtained from the instrument throughout most of the campaign. The reasons for this noisy signal remain unknown, but it is speculated that resulted from the sensitivity of the C band H-pol to RFI and cable connection issues.

The geophysical impact on the variability of the collected dataset was assessed by looking into the spatial and temporal variability of $T_b$ and comparing it with that of TIR SST. The ISAR instrument and the three usable channels (X band H- and V-pol, and C band V-pol) of the EMIRAD instrument were analysed. Figure 6 showed an overall higher variability for the PMW bands compared to the TIR SST. The H-pol channel exhibited the highest temporal and spatial variability for the moving data, as shown in Figures 6a and 6b, and this variability was even more pronounced in the port data (Figure 6c). However, for
sky measurements in which little geophysical impact is involved, the variability was minimal (Figure 5). This confirms what is already known from the literature regarding the impact of various physical parameters on $T_b$ (Nielsen-Englyst et al., 2021; Wentz and Meissner, 2000). The sensitivity of simulated $T_b$ to geophysical parameters for the EMIRAD frequencies was used to quantify the impact of wind speed on the H-pol channel (Figure 7a). Although wind speed and wind direction measurements were not directly available for this analysis, ERA5 data were used as a coarse approximation. This may explain the general
variability observed in the X band H-pol signal (lower panel in Figure 4), as wind conditions can change rapidly compared to other geophysical parameters that vary more slowly, such as SST. The uncertainty analysis revealed that a wind speed error of 2 m/s would result in a $T_b$ measurement error of 1.71 K for this particular channel, while for the V-polarization of both channels, the error would be below 0.5 K.

The forward model simulations also revealed a strong sensitivity of both vertical and horizontal polarization at the C and
X band frequencies to minor changes in the incident angle (Figure 7b). This is consistent with previous studies (Wentz and Meissner, 2000) and can be explained by the angular dependency of sea surface emissivity, which is largely described by the Fresnel equation, being greater in microwave regions than in infrared regions (Masuda et al., 1988).





The TIR SST$_{skin}$ measurements from the ISAR instrument were used as a baseline for deriving the PMW SST$_{subskin}$ retrieval coefficients from T$_b$ measurements obtained by the EMIRAD instrument. The PMW retrieval thereby implicitly involved adjusting for the mean sub-skin towards skin temperature, through the coefficient that incorporates a constant offset (c$_0$). The variability in the cool skin effect is, however, still present when comparing the different types of retrieved SSTs. Furthermore, wind speed is a primary driver of this effect and is a component of equation 1. As per Donlon et al. (2002), the cool skin effect tends to be smaller above a wind speed of 6 m/s, particularly at night. The wind speed dependence analysis of the SST differences for the examined dataset collected during daytime indicates that SST$_{subskin}$ is generally warmer compared to SST$_{skin}$ when the wind speed exceeds 6 m/s and cooler at lower wind speeds. However, the later output prevails ($\mu$ = -0.02 in Figure 10a) as daytime wind speeds were generally low. In contrast, the analysis for data collected at port did not take into account the wind speed in the retrieval process. Consequently, a relatively warmer skin temperature is observed during daytime, while lower skin temperature is noted at nighttime, aligning with finding in Donlon et al. (2002).

The regression analysis was conducted to define the retrieval equation, and its performance was assessed with and without splitting the dataset. Although the splitting process significantly reduced the dataset, the coefficients of determination ($R^2$) for the three fits changed by less than 0.01 for each data subset. However, it is important to note that the RMSE between the observed and retrieved SST increased by 0.17 K when considering the port data. On the other hand, when the regression was applied to the entire dataset without splitting, it resulted in a very small mean bias of the SST, raising concerns about potential over-fitting of the regression model. Thus, coefficients were obtained using the split dataset (training and test) despite the limited number of data matchups available.

The uncertainty analyses highlighted the sensitivity of the retrieval method to the incident angle. An error of 0.2 degrees in $\epsilon_\theta$ would have a larger impact on the V-polarization of both channels, resulting in a 0.39 K uncertainty in the estimation of SST. This emphasizes the importance of accurately measuring $\theta$ along with T$_b$ for the retrieval of SST$_{MW}$.

The results presented in this study were based on a dataset of small sample size; therefore, findings should be interpreted with caution. Nonetheless, they provide insights into the comparison between SST$_{subskin}$ and SST$_{skin}$. When comparing the retrieved PMW SST with measured TIR SST, the results indicate a general concordance between the two types of SSTs, which largely align within the derived uncertainty budget for the PMW SSTs, attributable to instrumental and geophysical factors. One exception is, however, during the last part of the campaign from Iceland to the Faroe Islands. These disagreements are likely due to the lack of precise observational data on the geophysical parameters that influence the signal variability (e.g. incident angle, wind speed and sea surface roughness).

## 7    Conclusions

In 2021, an unprecedented preliminary study was undertaken, marking a significant step forward in the field of oceanic temperature monitoring. This study involved the simultaneous acquisition of shipborne data utilizing both TIR and PMW instruments. These instruments were mounted in close proximity during a week-long campaign traversing from Denmark to Iceland. No-



tably, the PMW radiometers were specially refurbishment specifically for this study, while a well-documented TIR radiometer served as the reference for retrieving SST from the PMW measurements.

The analysis of the unique dataset obtained has yielded valuable insights into the intricate challenges associated with capturing and establishing the relationship between skin and sub-skin SST. This study underscores the pressing need for further advancements in PMW instrument design to ensure a robust association between these two SST observations.

Furthermore, our assessment of the uncertainty budget for the PMW observations included a sensitivity analyses of $T_b$ to various physical parameters, particularly emphasizing the importance of accurately accounting for the incident angle of PMW measurements as well as the wind speed and direction.

Drawing from the data collected and the knowledge gained from PMW brightness temperature measurements, this study proposes enhancements for the design and execution of future TIR-PMW shipborne/aereal inter-comparison campaigns:

1. Prioritize instrument design considerations: Special attention should be given to the instrument design, particularly in terms of its sensitivity to external RFI noise. In this study, the C band H-pol channel output data was affected due to high RFI levels, making it unusable. Therefore, measures should be taken to minimize RFI and optimize instrument performance.

2. Address cable losses: Account for changes in cable losses when manipulating the antennas, as this can have a noticeable impact on the performance of specific channels. For instance, in this case, the X band H-pol was affected. By addressing cable losses, the accuracy and reliability of the measurement can be improved.

3. Incorporate independent instrumentation for comprehensive data collection: To gain a deeper understanding of the effects of incident angles on PMW data collection, it is recommended to equip the PMW instruments with additional independent instrumentation. This should include instruments to measure geolocation, inertial measurement units, or other external parameters for each $T_b$ sample collected. This additional data will provide valuable context and improve the interpretability of the PMW measurements.

4. Consider complementary weather observations: To account for the sensitivities of PMW instruments to local atmospheric variations at small scales, it is advisable to ensure that the TIR-PMW matchup dataset encompasses complementary weather conditions throughout the ship's course. This will provide a broader range of conditions for analysis and enable a more comprehensive assessment of the instruments' performance.

5. In situ observations of $\text{SST}_{subskin}$s: For improved characterization of the PMW retrieval algorithm and its uncertainties, as well as evaluating the average cool skin effects, it is advised to equip the ship with instrumentation capable of monitoring in situ $\text{SST}_{subskin}$s throughout the cruise.

6. Ensure a larger matchup dataset: Because of the multiple conditions that prevent simultaneous data collection from different instruments, a longer campaign or a larger sampling rate of the collection will ensure a more confident conclusion about the retrieval algorithm's effectiveness and a more significant data comparison.



In implementing these recommendations, future TIR-PMW shipborne/aerial inter-comparison campaigns stand to benefit from enhanced instrument performance, improved measurement accuracy, and a more profound understanding of the intricate relationships between TIR and PMW measurements. This preliminary study serves as a pivotal milestone in laying the groundwork for simultaneous TIR-PMW observations, offering a unique opportunity to delve deeper into the distinct SST measurements captured by these methods. By addressing these key factors, researchers can build on the foundational insights from this study and move towards a more thorough understanding of oceanic temperature dynamics. Through collaboration and careful consideration of these recommendations, the scientific community can drive progress in oceanic temperature monitoring techniques. This advancement is crucial, especially in light of upcoming projects like CIMR, emphasizing the need for improved combined methods in SST monitoring. Such progress holds significant implications for climate research, environmental management, and maritime industries.



*Author contributions.* Jacob L Høyer (JLH), Sten S. Søbjerg (SSS) conceived the idea and design of the experiment. SSS and Sotirios Skarpalezos (SS) executed the campaign. Guisella Gacitúa (GG) processed and analysed the data. Hoyeon Shi (HS) performed the sensitivity and uncertainty analyses. A thorough review was made by Ioanna Karagali (IK) and Craig Donlon (CD) and all co-authors contributed to the interpretation of results.

*Competing interests.* The authors declare no conflict of interest.

*Acknowledgements.* The authors gratefully acknowledge the support of the ESA FRM4SST project (ships4sst.org), which provided funding for this research. We also thank the Japan Aerospace Exploration Agency (JAXA) for producing the AMSR2 level 2 data product, accessible through the Globe Portal System (G-Portal) at https://gportal.jaxa.jp/gpr/. The European Centre for Medium-Range Weather Forecasts (ECMWF) is acknowledged for providing the ERA5 data, which were instrumental to this study, particularly for obtaining wind speed information. Special thanks to Mads H. Ribergaard and Till S. Rasmussen from Ocean DMI for providing the DMI-HYCOM salinity dataset.

Furthermore, we express our gratitude to the Smyril Line passenger ferry, Norröna, for their invaluable support of this research endeavor.

455

460




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
