# Peer review of "Shipborne Comparison of Infrared and Passive Microwave Radiometers for Sea Surface Temperature Observations"

_EGUsphere, 2024_

## Author Comment (AC1)

**Response**

We thank the editor and reviewers for their time and effort in evaluating this manuscript and for their valuable recommendations. Based on these suggestions we have made changes to the text and included further analyses of the uncertainty estimation in the PMW SST. We believe that the manuscript has been substantially improved and our amendments will address the reviewer's concerns, making it suitable for publication.

Please find our response to your specific comments below (in blue) with reference to specific changes by Line number of the revised version.

**RC1**: 'Comment on egusphere-2024-542', Anonymous Referee #1, 25 Apr 2024

The manuscript by Gacitúa et al. describes a novel ship-based intercomparison of sea surface temperature measurements using infrared and microwave radiometers. This is, to my knowledge, the first evaluation of such a deployment and the work has implications for future validation of satellite SST measurements and intercomparison of skin and subskin measurements. As such, I believe the manuscript can be an important contribution to the literature after revisions that, in particular, help to place the use of the microwave radiometer in better context of all the potential validation approaches and more directly speak to the challenges associated with variations in the incidence angle. My suggested revisions are largely related to presentation and context, but could involve some additional computations associated with incidence angle variations.

The conclusions speak toward understanding the intricate relationships between TIR and PMW measurements. This is indeed an important end goal and justification for the type of work included in the study. To help get to that point, I think there could be more discussion up front and throughout on the overall context and approach to such studies. How will passive microwave radiometers be used and fit into an overall validation and research strategy? Given the measurement uncertainties involved (that are highlighted in this study), how suitable is a microwave radiometer for ultimate validation of subskin temperatures and differences from the skin temperature measured by an infrared radiometer?

We appreciate the reviewer's insightful comment and agree that it is essential to discuss the broader context and approach when comparing the two measurements. In response, we have clarified how the differences between the two measurements were considered in our analyses, with particular attention to the uncertainties involved.

We emphasize that the motivation of this study is more focused on supporting harmonizing satellite IR and PMW SST records through the knowledge and experience gained from the intercomparison campaign, than collecting reference measurements of sub-skin temperature to validate satellite PMW SST retrievals. Thus, a question such as 'Can we estimate PMW SST that is comparable to IR SST despite the differences in the characteristics of skin and sub-skin SSTs?" is addressed in this study. To make the motivation clearer to the readers, this is now more explicitly elaborated in the 5th paragraph of the Introduction. We modified the text in the Discussion to address the measurement discrepancies and to detail the limitations in comparing these two datasets. We have also outlined how these challenges were mitigated to the best of our ability using the available data and how future studies will benefit from the lessons learned.

While a skin temperature measurement cannot practically be obtained by other than radiometric approaches, the same is not necessarily true for the subskin temperature. Line 437 speaks to the

potential importance of a coincident measurement of the subskin temperature - this is indeed a critical point and I think this should be addressed up front.  The manuscript could touch on, in at least a cursory manner, the relative tradeoffs between different measurements – a physical measurement is potentially more accurate but it can be challenging to get at the precisely desired depth without disturbing the measurement.  When and for what purpose would you potentially want to use the different measurements?

We agree that this is a critical aspect and appreciate your suggestion to further elaborate on the different measurement techniques. To address your concerns, we have revised the discussion on the relative advantages and limitations of additional data collection to improve the comparison of these two measurements in the future (Last paragraph in Discussion section).

Moreover, I think somehow the paper could touch a bit more on the differences between validating the general form of the retrieval algorithm, vs specific measurements from the deployed radiometers.  I appreciate that these are closely intertwined, but the uncertainty analysis could potentially speak a bit more explicitly to these differences.  Are we really ready to directly evaluate the performance of a specific radiometer or are we still very early on in validating a concept?  If the ultimate goal were to validate a spaceborne microwave radiometer, would that be better done by looking at directly intercomparing the brightness temperature rather than the retrieved temperature, thereby eliminating uncertainty relative to the retrieval algorithm?  Managing expectations like this could help further establish the unique significance of the potential study.

We assess those changes made to the text in the Introduction, Discussion and Conclusion sections improved the clarification of the goals and limitation of this particular study.

Another more overarching point that could benefit from some increased treatment is the effect of sensor incidence angle.  Line 416 in the conclusions speaks to the presented assessment speaking to the importance of accurately accounting for the incidence angle, but I don't feel this is done to the potential degree required.  There is some discussion near line 396 but this is significant and could use more up front and overall discussion.

This point was addressed by, firstly, introducing the concept of incidence angle, and its relevance in microwave and infrared radiometry, in the second paragraph of section 4.3. Secondly, highlighting the output of the sensitivity uncertainty analyses (Line 431) to give context to the recommendation of additional measurements .

The error of 0.2 discussed is actually smaller than some of the variations observed in practice.  The manuscript states (near line 230) that effects based on the IR angle were explored but not utilized.  How did the results compare?

The sentence in 230 is removed as it is not relevant to discuss in depth the initial approaches to the final retrieval method. We instead slightly modified the text in section 5.1 to emphasize how the retrieval equation was defined, i.e. as a result of the forward model output and data availability.

 While the angle measured by the infrared sensor obviously can't be used directly, was the offset in incidence angles a constant that could possibly be accounted for?  Additionally, the results are presented in terms of the standard deviation of the incidence angle apparently computed over the entire deployment but in Fig 9 it is clear that over some periods the variation is much greater.  What is the expected uncertainty under these specific periods?  Is there some potential correlation between the observed errors/differences and these periods with greater angular variations? (the

results in the figure might suggest that there is). Line 416 in the conclusions speaks to the presented assessment speaking to the importance of accurately accounting for the incidence angle, but I don't feel this is done to the potential degree required.

Figure 9 shows the standard deviation of the roll angle (based on a window of 10 samples) measured by the IR radiometer (Line 189). This time series values were used to account for $\varepsilon_\theta$ included in the uncertainty budget (Line 294). The implication of a higher standard deviation over some periods is now accounted as the total uncertainty shown as "error bars" added to Figure 9 (instead of a constant average uncertainty).

To summarize, because the absolute incidence angle could not be considered in the regression equation due to the lack of appropriate observations, its uncertainty was accounted for in the regression and the uncertainty calculation steps. We recommend that the incidence angle is utilized more explicitly in future studies.

Other minor comments

Line 9 – "stable": While I understand what is meant here, I wonder if another word might be used in place of "stable" as it also often is used to refer to atmospheric conditions related to the slope of the temperature profile.

Re-phrased to avoid using the concept of "stable".

Line 12: Since the previous sentence speaks of both skin and subskin temperature, I think it is important to be clear as to what quantity (presumably the microwave) the error values refer to.

Added specification of the error "in the retrieved PMW SST"

Lines 24-25, "brightness temperature…emitted from": "representative of" might be more appropriate that "emitted from" as radiation is "emitted" but not brightness temperature.

Changed to "brightness temperature … representative of thermal emission from the subskin layer ..."

Line 43: O'carroll -> O'Carroll

Corrected

Line 51: Isn't what is novel here the use of the passive microwave radiometer? Thermal IR instruments have been deployed extensively in the past.

Rephrased

Section 2.1: I think it is important to mention that the ISAR is just one such instrument. Other instruments provide radiometric measurements of the skin temperature including even more sophisticated instruments like the M-AERI. It could be useful to cite previous studies showing positive intercomparison results between other IR radi  ometers.

Intercomparisons between FRM TIR radiometers were already cited in Line 45.

Line 80: The Hoyer reference could be improved. From the listed website there are further directories and then multiple papers which are hard to align with what is referred to here. The most seemingly closely related effort is not available through this website. Is there any other information on this radiometer available?

Links to the documents are now updated in the reference list.

Line 90:  While there is a reference here, given the importance of these steps, it might be helpful to the reader to explicitly say a bit more about the different steps.

The four steps are now mentioned more specifically.

Line 93:  Range of datasets "was" -> were.  Also line 97, data was -> data were

Corrected

Line 95:  SLSTR definition later (line 106) should be moved to this first use.  What is the source of these data?  I see acknowledgement and reference of other data sets but I might have missed the source here.

Corrected

Line 136:  Is there any possible direct calibration possible for the microwave brightness temperature/radiometric measurement?  Is there any corresponding absolute calibration?

We did not perform an absolute calibration of our PMW instrument during the campaign. Although absolute calibration can theoretically be achieved by pointing the antenna towards the sky, this method is not valid in a shipborne context due to the atmospheric influence on the measurements. Unlike in spaceborne scenarios, where pointing to the cold sky can provide a reliable calibration point, the atmospheric effects encountered during the campaign prevent this approach from serving as an accurate calibration reference.  Therefore, it could have been ideal to employ an alternative method for the absolution calibration; however, it was not available by the time of the campaign. As of now, a more detailed description of the calibration procedure of the EMIRAD was added (Line 95).

Line 199:  "raw clean" seems to me to be a bit of a misnomer – instead perhaps "raw data with XX removed"

Corrected

Section 4.2:  I wonder if it would be worth commenting on the potential use of detrending prior to the running average.  While the sky variability shown in Fig. 5 is longer term, is there any potential influence over the longer 60-min averaging time?

In Section 4.2, the analysis focuses on the variability of the data excluding the sky measurements, and no detrending was applied prior to the running average. We believe that the current approach, using the running average without detrending, is sufficient for the purpose of comparing the variability across all radiometers. While detrending could be useful in some contexts, the goal here was to capture the overall variability patterns, and the 60-minute averaging period is appropriate for that scope. Therefore, we did not find it necessary to apply detrending in this case.

Line 219 and prior:  What is the effective emissivity at these microwave frequencies?  I believe there are significant differences from 1.  Given the longer-term variation of the sky temperature shown in Fig. 5, is there any potential for the influence of reflective effects?  How are these handled?  If the microwave data are independent of cloud liquid water and water vapor, what is the source of the longer-term sky variation in Figure 5?

The C- and X-band sea surface emissivity at a 55 degree incidence angle is approximately 0.55 for vertical polarization and 0.25 for horizontal polarization. This implies that 45% of the downwelling atmospheric emission can reach the sensor for vertical polarization, and 75% for horizontal polarization, after being reflected at the sea surface. The typical range of downwelling radiation Tb is 3-5 K. Therefore, as it is pointed out, surface reflected sky radiation can influence the observed brightness temperature by about 1-1.5 K, particularly for the horizontal polarization.

One of the purposes of having sky measurements was to measure atmospheric downwelling radiation to account for the above-mentioned effect. Since the sky measurements at C and X bands should consist of insignificant atmospheric emissions and cold cosmic microwave background (CMB, approximately 2.7 K), the Tb measurements are expected to be below 5 K. However, as shown in Figure 5, it was not as cold as 5 K but shows positive offsets. As there is not enough information to determine the exact cause, we assumed that despite the positive offset in these sky observation, its variability appropriately conveys the geophysical variability (i.e., changes in the sky condition) and the instrument's random noise.

To address your concerns, we provide a detailed explanation of this matter in the revised manuscript to provide context (Line 150), address the potential influence of reflective effects in the uncertainty budget estimation (Line 326), and suggest improved data availability in future studies and campaigns (Line 483).

Fig 5b: Given the occurrence of greater incidence angle variations, is it worth expanding the x-axis here to account for the full range of variations seen?

We assume that the reviewer means Fig 7b that shows Tb due to incidence angle deviation as a result of the forward model. We believe that given the resulting curve, which looks close to linear, one can easily estimate the approximate value of Tb for higher values of incidence angle deviation.

Line 230 and nearby: As touched on in my more general comments, could a constant offset in angle be applied? Given the large sensitivity to incidence angle, I believe more justification of why this approach is sufficient is required earlier on.

We did not use a constant offset but the standard deviation of the platform roll (measured by the IR radiometer) which is based in a window of 10 samples. We now specify this in section 5.1 for clarity and the uncertainty budget now includes the uncertainty in the incidence angle for each point in the dataset.

Line 248: I appreciate the limitation associated with the quality of the ERA5 winds, but I am less clear on use of the roll angle to justify setting winds to zero. Doesn't wind have an effect on roughness that can affect the measured brightness temperature?

Yes, wind speed represents the surface roughness. Accordingly, setting the wind speed to zero means considering smooth surface conditions. At port, changes in the roll angle were mainly caused by the surface waves (roughness). Therefore, we used the variability of roll angle as a measure of surface roughness for the port condition. As stated in line 248 in the previous manuscript, we decided to set wind speed to zero considering the measured roll angle for the two periods at port. The mean roll angle measured was 0.14 with a maximum value of 0.5, whereas the mean standard deviation is 0.0006 indicating very low roughness. We clarified this in Line 276 of the revised manuscript.

Table 3:  Tying in to a broader discussion on incidence angle, given the range of incidence angle variations shown in Figure 9, for instance, is a value of 0.2 degrees for the incidence angle sensitivity truly sufficient?  How are temporal variations accounted for?

Figure 9 has been updated to account for temporal variation in the uncertainty budget.

Line 330:  The difference here, as I understand it, is near 0.3 K – is this really "not significant"?

This has been rephrased.

Line 388:  The grammar of this sentence could be improved.

Corrected

Line 396:  Again tied to a potential broader discussion of incidence angle – Fig 9 suggests even larger variations in the incidence angle than 0.2.  How much larger does the effect get for these larger angular variations?

Figure 9 now shows temporally varying incidence angle uncertainty, accounting for small and large incidence angle uncertainties.

Line 410:  specially refurbishment -> specially refurbished?

Corrected

**RC2**: 'Comment on egusphere-2024-542', Anonymous Referee #2, 19 Aug 2024

Line 51: Not sure I understand why this is a first use of FRM TIR instruments as IR radiometers have been used for a long time and after the FRM4ST project many will have been checked against the SI. Do you mean the first FRM TIR against MW? Please make clear what you mean.

Rephrased to make sure it is understood that the uniqueness of the study is the comparison of simultaneous observation from PMW and FRM IR in close proximity to the sea surface (Line 53).

Line 76: EMIRAD instrument: Any comment regarding possible sideline contamination?

Sideline contamination is indirectly addressed in section 2.3 of the Characterization Report cited (Høyer et. al., 2021b). The antennas have not been characterized in a radio anechoic chamber, but Figure 2-5 in combination with Table 2-1 (on the report) provide theoretically expected antenna patterns. The patterns suggest that the antenna gain will roll off towards 90 degrees from bore sight, but they also demonstrate a relatively wide angular interval from which radiation is picked up. Contamination from sources near the horizon cannot be excluded, while the vessel itself is behind the antennas, where - in theory - no radiation is received. However - it is important to emphasize that the curves only show theoretical patterns and not measured ones.

We have now added text to section 2.2 on the manuscript to mention the theoretical approach taken with reference to the Characterization Report.

Please use uncertainty and error appropriately (see, for example, "the guide to the expression of uncertainty in measurement" for definitions). Examples:

Figure 5: Should be "instrument uncertainty" not "instrument error"

We have revised the concept of uncertainty and changed the use accordingly.

Line 180: Section 4.1. This section derive the MW instrument noise but there is nowhere where the instrument systematic uncertainties are discussed. Are these available anywhere? They will, of course, increase the total MW uncertainties.

Due to the lack of detailed information on the systematic uncertainties associated with the instrument, we have assumed that these uncertainties are negligible for the purposes of this study.

We recognize that systematic uncertainties could potentially increase the overall PMW uncertainties. However, without specific data or references available to quantify these uncertainties, we opted not to include them in our analysis.

To address this limitation, we have added a note in the revised manuscript acknowledging the absence of systematic uncertainty data and its potential implications on our findings (Line 337).

Line 188, 194, 196: Should be "instrument(al) uncertainty"

Changed

Line 238: Observation uncertainty not error

Changed

Line 250: What fitting process was used (LSQ?)

We used LinearRegression (from the Scikit-learn library), which is based on a weighted least squares (WLS) fitting process, with weights defined by eq. 2 in the manuscript.

Line 300: Section 5.3: Are there correlations between the MW/IR SST differences and elements of the retrieval function e.g. wind speed, incidence angle etc.?

Correlation between SST biases and elements of the retrieval function were looked into but no significant correlation was found. We opted not to include this analysis in the manuscript due to the small sample size, which we have discussed in the manuscript (e.g. Line 406, 447). However, we had performed a bias analysis across different wind speed ranges. This section has been revised to better present our findings (Line 438).